# Two Cases of 6-Pyruvoyl Tetrahydropterin Synthase Deficiency: Case Report and Literature Review

**DOI:** 10.3390/children10040727

**Published:** 2023-04-14

**Authors:** Lucia Maria Sur, Monica Alina Mager, Alexandru-Cristian Bolunduţ, Adrian-Pavel Trifa, Dana Teodora Anton-Păduraru

**Affiliations:** 1Faculty of General Medicine, University of Medicine and Pharmacy Iuliu Haţieganu Cluj-Napoca, 400015 Cluj-Napoca, Romania; 2Children’s Emergency Hospital, Motilor Street No 68, 400015 Cluj-Napoca, Romania; 3Faculty of Medicine, Medical Genetics, University of Medicine and Pharmacy Victor Babes Timisoara, 400349 Cluj-Napoca, Romania; 4Faculty of Medicine, Mother and Child Discipline, Department of Pediatrics, University of Medicine and Pharmacy Grigore T. Popa Iasi, 700115 Iasi, Romania

**Keywords:** 6-PTPSD, BH4Ds, HPA, NBS, pterins, late diagnosis, late treatment, neurotransmitters

## Abstract

6-pyruvoyl tetrahydropterin synthase deficiency (PTPSD) is a rare neurometabolic disease that can be diagnosed in newborn screening (NBS) and is part of the family of tetrahydrobiopterin deficiency disorders (BH4Ds). It is essential to diagnose and treat this disease early to prevent permanent neurological damage secondary to this neurotransmitter disorder. We present the first two cases of PTPSD in Romania that were genetically confirmed and treated late. Improving the diagnosis and monitoring procedures in Romania with correct metabolic management will prevent severe neurological impairment from PTPSD or other BH4Ds.

## 1. Introduction

6-pyruvoyl tetrahydropterin synthase deficiency (PTPSD) is a rare neurometabolic disease that can be diagnosed in newborn screening (NBS) and is part of the family of tetrahydrobiopterin deficiency disorders (BH4Ds) [1,2,3]. Because tetrahydrobiopterin (BH4) is a coenzyme for phenylalanine hydroxylase (PAH), tyrosine hydroxylase (TH), and tryptophan hydroxylase (TH), the deficiency will lead to hyperphenylalaninemia (HPA) and reduction in the synthesis of neurotransmitters in the brain (dopamine and serotonin) resulting in neurological abnormalities and intellectual impairment [4,5,6,7,8,9]. (Figure 1).

PTPSD has clinical manifestations in early childhood that include neurological signs (developmental delay, seizures, movement disorders, parkinsonism) and hyperphenylalaninemia (HPA) [10,11].

In Caucasians, the prevalence of BH4 deficiency is 1–2% of all HPA [1,9], and PTPSD is the most common cause of BH4 deficiency [3,6,9].

The incidence of HPA in Europe is approximately 1:10.000, and BH4Ds are 1–2% of all cases [9]. PTPSD is the most frequent of all BH4Ds [3,4,12].

It is essential to early diagnose children with PTPSD to prevent them from developing symptoms. After HPA is detected in NBS by dried blood spot (DBS) testing, the next step is pterin analysis and CSF studies [3,13,14,15].

We present the first two cases of PTPSD in Romania that were genetically confirmed and treated late.

Improving the diagnosis and monitoring procedures in Romania with correct metabolic management will prevent severe neurological impairment from PTPSD or other BH4Ds.

## 2. Case Presentation

Patient 1 is a girl diagnosed late, at nine months of age.

She first presented at our hospital at two months due to a positive newborn screening of HPA. Her first value on DBS was 331 µmol/L (normal value (NV) < 126 µmol/L).

She was born to consanguineous parents (both parents and grandparents are cousins). Her family history revealed that an aunt died in early childhood because of an unknown neurological disease. She is the second child of the family and she was born at 38 weeks of gestation (WOG), from a natural birth; her birth weight was 2300 g (1st percentile) and her length was 48 cm (27th percentile). The clinical exam was normal, except for a small umbilical hernia.

The serum phenylalanine (Phe) was 237.77 µmol/L (NV36.3–66.55 µmol/L). We then determined pterins on DBS, revealing a low biopterin concentration, indicative of PTPSD. We also determined the dihydropteridine reductase (DHPR) enzyme activity on DBS to exclude dihydropteridine reductase deficiency (DHPRD), which was normal.

The family did not bring her for a follow-up for seven months. At 9 months, she returned to our clinic because of hypotonia and neurological developmental delay (she was not holding her head from 6 months of age, was not sitting, and was not doing side turns).

We determined the serum Phe level, which was slightly elevated (75.6 µmol/L), and the patient still had low biopterin on DBS, and the neopterin/biopterin ratio was low (Figure 2).

The first neurological exam performed at two months was normal. The head circumference was within the normal range for her age. At the second admission, at nine months of age, she had severe hypotonia, poor head control, progressive motor impairment (loss of head control at six months), and mild hypokinesia. The patient also experienced oculogyric crises, interpreted as dystonia, dystonic postures in the upper limbs, and tremors triggered by psychomotor agitation (unintentional motions of the hands and feet, irritability, and restlessness) She also encountered a daily fluctuation of movement symptoms (worsening over the day, with improvement after rest). We performed an electroencephalogram (EEG) to exclude other neurological diseases, showing a regular awake pattern without epileptiform activity. At 11 months, she had a 48 h BH4 loading test that lowered the Phe levels to 53% after administration of Sapropterin dihydrochloride (10 mg/kg BW/day).

We then introduced treatment with L-Dopa/Carbidopa, starting at 0.5 mg/kg BW/day, and increasing to the current very low dosage of 2.5 mg/kg BW/day.

We could not establish the severity form of PTPSD because we could not perform CSF studies. The administration of neurotransmitters was based on the clinical response.

At the age of 1 year, we performed genetic testing, and a hyperphenylalaninemia NGS (next generation sequencing) panel (genes analyzed: *DNAJC12, GCH1, PAH, PCBD1, PTS, QDPR, SLC25A13, SPR*) showed a homozygous genotype for the pathogenic c.84-3C > G variant of the *PTS* gene.

She started treatment with Sapropterin dihydrochloride at one year and four months (there was a delay in treatment because of the absence of a standard protocol for BH4D therapy in Romania). The blood phenylalanine concentrations decreased after BH4 supplementation, and the patient had normal plasma phenylalanine levels and regular protein intake.

We tried two types of 5-hydroxytryptophan (5-HTP) treatment, but the patient did not tolerate it because of gastrointestinal adverse reactions; she had vomiting and diarrhea, and we had to stop the treatment.

After the start of specific treatment (Sapropterin dihydrochloride and L-Dopa/Carbidopa), the evolution indicated a progressive recovery of motor skills, improved head control and sitting position without support. The hypotonia, hypokinesia, and dystonia were ameliorated. She had mild cognitive impairment, with a developmental quotient (DQ) using the Portage Assessment Scale of 54 at the first assessment, which decreased to a DQ of 45 (moderate cognitive impairment) without treatment and increased to a DQ of 58 after the specific treatment was started. She also had kinesiotherapy almost every month since the diagnosis, and she walked at two years with unilateral support and spoke a couple of meaningful words.

To exclude other neurological diseases, we performed cranial magnetic resonance imaging (cMRI) at one year and four months, which was normal.

As a less expensive alternative for clinical monitoring, we determined blood prolactin serum levels, which were high (dopamine inhibits prolactin, correlating with low dopamine levels), meaning that the metabolic disease was not adequately controlled.

Patient 2 is a boy, also born from consanguineous parents (third cousins), from a pathological and incorrectly monitored pregnancy, with the threat of miscarriage, born with c-section, with an average birth weight (3030 g, 46th percentile). The clinical exam indicated facial dysmorphism (low inserted ears, short neck), seborrheic dermatitis, heart murmur, ankyloglossia, and generalized hypotonia.

The first value of Phe at NBS was 348 µmol/L, and the serum Phe levels at six weeks were 312.4 µmol/L (NV 36.3–66.5 µmol/L). The DBS attested a high neopterin and low pterin ratio, indicating PTPSD. DHPR enzyme activity in the DBS was normal.

The heart ultrasound detected a patent foramen ovale; the abdominal ultrasound revealed minimal right hydronephrosis; and the head ultrasound normal, ophthalmological consult were normal.

The routine lab tests indicated hypercholesterolemia, positive COVID IgG antibodies, hyperammonemia, positive IgG anti-CMV antibodies, slightly elevated liver enzymes, and iron deficiency anemia.

At the age of four months, we collected DNA from peripheral blood cells to conduct genetic testing, and a hyperphenylalaninemia NGS panel was performed revealing the same mutation as the first patient, the c.84-3C > G variant of the *PTS* gene, in homozygous state; this could be due to the patients belonging to the same ethnicity and the same geographical area.

At the age of 5 months, our patient’s cerebrospinal fluid (CSF) was analyzed, and neurotransmitter metabolites 5-hydroxindolacetic acid (5-HIAA) and homovanillic acid (HVA) were measured to determine disease severity. The patient had low HVA and 5HIIA in the CSF, so we could include the patient in the severe form of PTPSD.

We did not perform a BH4 loading test. Still, he received donation treatment at five months with Sapropterin dihydrochloride (10 mg/kg BW/day) until he was included in the National Treatment Program.

The therapy with Sapropterin dihydrochloride was discontinued because of the latency period until the introduction into the National Treatment Program at eight months. We also introduced L-Dopa/Carbidopa treatment (0.5 mg/kg BW/day).

An initial neurological assessment was performed at two months for the second patient. The head circumference was within the normal range for his age. The patient had mild axial hypotonia with asymmetric postures. No movement disorders were reported at the first assessment, but they occurred in the upper limbs by nine months of age.

An electroencephalogram was performed, and no epileptic activity was recorded during the tracing. The developmental quotient (DQ) was 90, indicating that the patient had moderate cognitive development.

At the age of 10 months, after one month from the start of the specific treatment, the hypotonia and dystonic postures had significantly improved, and he could stand without support. Fine motor development and cognitive development were within normal limits. We also reported excessive salivation since the age of 5 months, that improved after treatment.

We determined plasma prolactin levels for this patient, which were still higher than the normal range, indicating that we still had not reached the proper dosage of neurotransmitters to control the disease.

Phe levels on follow-up were within the normal range, and he was not on a Phe-reduced diet.

Regarding the treatment, both patients received Sapropterin dihydrochloride (10 mg/kg BW/day) and L-Dopa/Carbidopa (dose range 2 mg/kg BW/day), with an increasing dose over the next months. None of the patients is taking 5-HTP because of the gastrointestinal adverse reactions; both had vomiting and diarrhea and we had to stop the treatment.

Both children have developmental delays, and treatment was introduced late (at one year and four months for patient 1 and at eight months for patient 2).

Follow-up was done every three months since Sapropterin dihydrochloride and L-Dopa treatments were introduced.

The data about the demographic, biochemical, and genetic descriptions of our two patients are available in Table 1.

## 3. Discussion

In our case presentation, consanguinity was reported in both of our patients. Similarly, Almannai et al. reported consanguinity rates as high as 89% in Arab subjects, mostly from Saudi Arabia. Carducci et al. reported a 57% rate in Jordanian patients. Bozaci et al. reported 77% consanguinity in Turkish patients. Blau et al. concluded that the highest incidence of PTPSD in Saudi Arabia is probably due to the high consanguinity rate [15,16,17,18].

As cited in the literature, our patients had neurological and cognitive impairments. Most patients with BH4 deficiency have hypotonia, impaired motor and cognitive development, movement disorders, parkinsonism, and rest tremors [3,19].

Truncal hypotonia, postural instability, dystonic limb movements, hypersalivation, and oculogyric crises were reported in a long-term follow-up of Chinese patients who received delayed treatment for PTPSD, as was also seen in our two patients that also received late treatment. Oculogyris crises were reported in 5–15% of PTPSD patients, and were observed in patient 1 [3,19].

One of our patients reported excessive salivation. Lee et al. identified hypersalivation in 40% of their 10 PTPSD patients [20].

Our two patients revealed improvements in the DQ (developmental quotient) scale scores after receiving neurotransmitter treatment, as cited in the literature. Wang et al. indicated that language, adaptability, and at a later age, mathematics were fragile areas for PTPSD patients. Lee et al. also used DQ as a tool to establish neuropsychological impairment. They observed improvement after administering neurotransmitters in the 15 years of follow-up of 10 PTPSD patients, even in severe forms [20,21].

One of our patients had intense irritability. Only 10% of patients with PTPSD have reported this symptom in the literature [3,16].

Prematurity was not found in our patients [3,16], but one had a low birth weight. However, according to the literature, microcephaly is more common in PTPSD. None of the Romanian patients with PTPSD had microcephaly [20,22].

The pattern of pterins in PTPSD in DBS and urine consists of highly elevated neopterin and low biopterin, similar to our patients [3,13].

We selected the DBS sample for measuring pterins because our hospital does not have the instrumentation to perform the measurement of pterins in urine. Transportation to an external laboratory was not the best option because the pterins in urine are more susceptible to degradation by light and high temperatures [13].

Levels of 5-hydroxyindolacetic acid (5-HIAA) and homovanillic acid (HVA) in CSF are usually low in BH4Ds; our second patient had his CSF analyzed, and we could include him in a severe phenotype of PTPSD [3,5,9,10].

We elected serum prolactin as a marker to evaluate the L-Dopa dose. Dopamine is an inhibitor of prolactin secretion, and serum prolactin can be elevated in dopamine deficit disorders. The serum prolactin was elevated in both patients, indicating that we must increase the L-Dopa/Carbidopa dosage [22,23].

Ogawa et al. described a case of PTPSD with a more significant correlation of L-Dopa dosage with serum prolactin levels than CSF homovanillic acid levels [24].

Mutation detection is the preferred method for diagnosis [8,24].

The c.84-3C > G variant is considered pathogenic in the ClinVar database (Variation ID: 553378), and was previously reported to be associated with as a severe form of PTPSD in Albania, Italy, and Iran [9,18,25,26].

Patients with PTPSD may have abnormal brain imaging results such as delayed myelination, periventricular hyperintensities, and brain atrophy. We performed a brain MRI for one of our patients, and we did not find hyperintense lesions in the periventricular white matter [9,19,21,27].

Patient 1 had a BH4 loading test. As there is no uniform procedure available, we opted for the 48 h test with a dosage of 10 mg/kg BW/day, and the levels of Phe were lowered by 53% [3,28,29].

Sapropterin dihydrochloride treatment was preferred instead of the Phe-reduced diet to normalize the levels of Phe, first because the studies revealed that this is an easier way to monitor Phe intake and second because of lack of cooperation for a Phe-reduced diet regarding our patients’ families [18].

The combined treatment with Sapropterin dihydrochloride and L-DOPA/Decarboxylase inhibitor improved the clinical manifestations of our patients, consistent with the results of other studies [5,30,31,32].

We were not inclined toward a monotherapy of Sapropterin dihydrochloride because our patients had developmental delays and clinical neurological signs. Our patients had no adverse reactions secondary to Sapropterin dihydrochloride administration [31].

We used an L-DOPA/Carbidopa 10:1 preparation with an extemporaneous prescription because of our country’s lack of suspension treatments and since the literature did not report any side effects compared to the 4:1 preparation. Our two patients did not reach the target treatment dose of L-Dopa/Carbidopa of 10 mg/kg BW/day [3,33].

A study indicating a discontinuity of 5-HTP treatment registered no signs of neurological deterioration. Our two patients did not tolerate the 5-HTP treatment provided in Romania because of gastrointestinal side effects (vomiting and diarrhea) [3,34].

Patients from Taiwan with PTPSD discontinued 5-HTP treatment for 15 years and have remained free of recurrent neurological or psychiatric symptoms [1].

The long-term outcome of the disease is influenced by the time of the initiation of treatment [3,35,36,37].

Kao et al. mentioned that even PTPSD patients treated early have subtle neurological dysfunctions [38].

## 4. Conclusions

PTPSD is rarely diagnosed in our country because of the absence of a standard evaluation protocol after identifying HPA during NBS. Just two centers in Romania are evaluating pterins on DBS to investigate BH4Ds further.

## Figures and Tables

**Figure 1 children-10-00727-f001:**
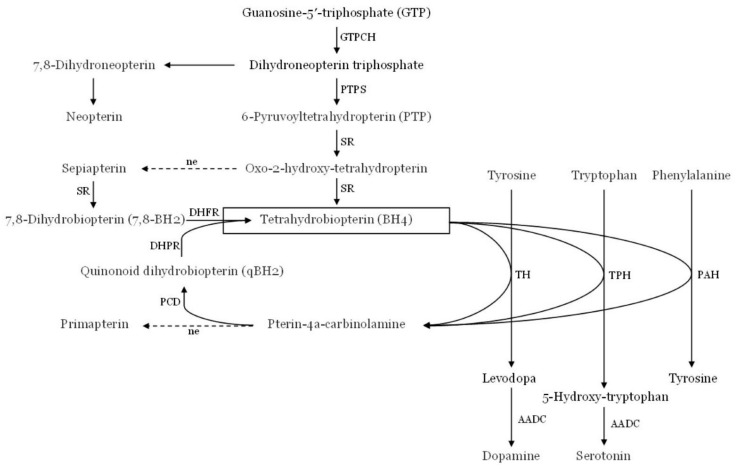
The metabolic pathway of tetrahydrobiopterin (BH4). AADC, aromatic amino acid decarboxylase; DHFR, dihydrofolate reductase; DHPR, dihydropteridine reductase; GTPCH, GTP cyclohydrolase 1; ne, non-enzymatic; PAH, phenylalanine hydroxylase; PCD, pterin-4a-carbinolamin dehydratase; PTPS, 6-pyruvoyltetrahydropterin synthase; SR, sepiapterin reductase; TH, tyrosine hydroxylase; TPH, tryptophan hydroxylase (adapted from Ref. [3], Figure 1).

**Figure 2 children-10-00727-f002:**
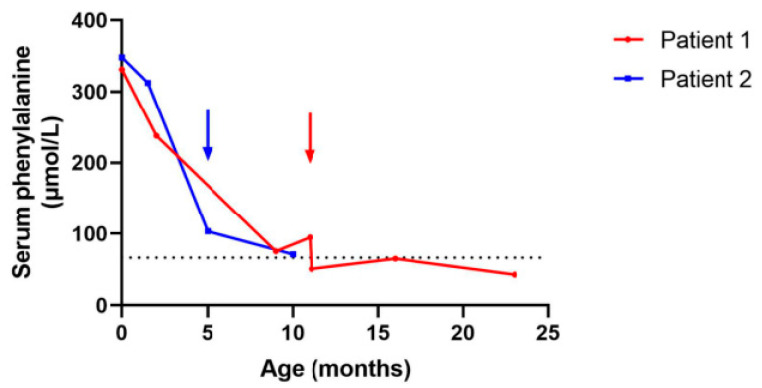
Dynamic representation of the phenylalanine levels in our patients. The dotted line represents the normal upper limit of serum phenylalanine concentrations (66.5 µmol/L). The arrows represent the start of the specific treatment (Sapropterin).

**Table 1 children-10-00727-t001:** Demographic, biochemical, and genetic description of our patients.

	Patient 1	Patient 2	Reference Values
Sex	Female	Male	
Birth weight (g)	2300	3030	
Birth length (cm)	48	?	
Birth gestational age (weeks)	38	?	
Consanguineous parents	Yes	Yes	
Family history	Yes (metabolic and neurologic—no specification)	No	
Positive newborn screening	Yes	Yes	
Age at diagnosis (months)	9	4	
Age at initiation of treatment (months)	11	5	
Screening phenylalanine level (µmol/L)	331	348	<126
Initial phenylalanine level (µmol/L)	237.7	312.4	36.3–66.5
Neopterin (nmol/g Hb)	2.29	4.52	0.19–2.93
Biopterin (nmol/g Hb)	0.03	0.12	0.08–1.20
Pterin ratio	1	3	16.0–64.7
DHPR activity (mU/mg Hb)	2.5	2	>1.1
Prolactin (ng/mL)	31.35	36.43	<30
PTPS mutation	c.84-3C > G (homozygote)	c.84-3C > G (homozygote)	

? no data.

## Data Availability

Not applicable.

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
