# Peer review of "Two Cases of 6-Pyruvoyl Tetrahydropterin Synthase Deficiency: Case Report and Literature Review"

_children, 2023, doi:10.3390/children10040727_

Round 1

Reviewer 1 Report

Two Cases of 6-Pyruvoil Tetrahydropterin Synthase Deficiency: Case Report and Literature Review

The described case report is interesting and due to the rarity of PTPS it is instructive, but some statements or sentences need to be improved.

The reviewer thinks that the subject is suitable for the Children journal. My minor comments and suggestions are shown below.

1. In the first case, when the biopterin levels were found to be low, was genetic testing done? Why did the authors stop here when they knew they were most likely dealing with a pterin problem? Did they try to find them and properly enlighten the family about the consequences?

2. In the second case, HVA and 5-HIAA levels were measured from CSF. Please write the exact values and the corresponding normal ranges in the main text. Was it possible to measure 3-O-methyldopa and 5-HTP levels from CSF? If so, what were the results?

3. In the second case the authors did not perform BH4 loading test because of financial problems. Have you considered DOPA/CARBIDOPA and 5-HTP treatment without BH4? And if not, what was the reason?

4. What could be the biochemical explanation for the failure of 5-HTP therapy?

5. Table 1: The phenylalanine levels are in mg/dl while in the main text authors refer as μmol/L. Please unify it.

6. Abbreviations are incorrect in several places, and not all cases indicate what the abbreviation covers. Please correct the following:

- Line 20: 6-PTPSD usually PTPSd. Please correct it.  

- Line 22: BH4: the first time you use it, please write down what it stands for.

- Line 45: NV: the first time you use it, please write down what it stands for.

- Line 49: WOG: the first time you use it, please write down what it stands for.

- Line 90 and 91: DQ or QD? The first time you use it, please write down what it stands for.

- Line 144: not HTP, 5-HTP.

- Lines 186 and 194: not homovanilic acid, homovanillic acid.

7. Figure 1:

- Not Levo-dopa, correctly it is Levodopa

- Not AAADC, correctly the enzyme is AADC.

8. Grafic 1:

The y-axis is not linear. 0.5 and 1 are at the same distance as 4 and 8. Please fix it.

9. Moderate English changes required, please check the manuscript again.

Author Response

Thank you 

Reviewer 2 Report

With great interest, I have read the manuscript of Sur et al. entitled “Two Cases of 6-Pyruvoil Tetrahydropterin Synthase Deficiency: Case Report and Literature Review”.

In this study, the authors present the first two cases of PTPSD in Romania that were genetically confirmed and late treated.

I believe that the manuscript can strengthen it by providing additional information on the following issue:

The primary issues that need to be addressed include 1) organization and presentation of findings, and 2) readability.

A number of grammatical issues hinder readability. 

The title reflects well the contents of the paper.

Page 2

Line 54

The first time you use the term “DHPR”, put the acronym in parentheses after the full term

Line 66

What do you mean by “psychomotor agitation” in a patient at nine months of age?

Can be clarified?

Lines 68-69

I would suggest reworking this section. 

“She did not have any seizures, and the electroencephalogram excluded any anomalies”.

Line 89-91

Age at any assessment should be added between parentheses.

What kind of tool has been used to assess DQ?

Line 90 and 91

You should write DQ and not QD

Page 3

Lines 96-98

I would suggest reworking this section. 

Line 115

You must use italics for gene PTS

Line 116

“country” and not “county”

Line 118

5-HIAA and not ”HIIA”

Line 130-131

I would suggest reworking this section. 

Table. 1 and Grafic 1

The author should convert the Phe value from mg/dl to µmol/L.

Author Response

Thank you.
